# Conditional Random Fields via Univariate Exponential Families

**Eunho Yang**
Department of Computer Science
University of Texas at Austin
eunho@cs.utexas.edu

**Pradeep Ravikumar**
Department of Computer Science
University of Texas at Austin
pradeepr@cs.utexas.edu

**Genevera I. Allen**
Department of Statistics and
Electrical & Computer Engineering
Rice University
gallen@rice.edu

**Zhandong Liu**
Department of Pediatrics-Neurology
Baylor College of Medicine
zhandonl@bcm.edu

## Abstract

Conditional random fields, which model the distribution of a multivariate response conditioned on a set of covariates using undirected graphs, are widely used in a variety of multivariate prediction applications. Popular instances of this class of models, such as categorical-discrete CRFs, Ising CRFs, and conditional Gaussian based CRFs, are not well suited to the varied types of response variables in many applications, including count-valued responses. We thus introduce a *novel subclass of CRFs*, derived by imposing node-wise conditional distributions of response variables conditioned on the rest of the responses and the covariates as arising from univariate exponential families. This allows us to derive novel multivariate CRFs given any univariate exponential distribution, including the Poisson, negative binomial, and exponential distributions. Also in particular, it addresses the common CRF problem of specifying "feature" functions determining the interactions between response variables and covariates. We develop a class of tractable penalized $M$-estimators to learn these CRF distributions from data, as well as a unified sparsistency analysis for this general class of CRFs showing exact structure recovery can be achieved with high probability.

## 1  Introduction

Conditional random fields (CRFs) are a popular class of models that combine the advantages of discriminative modeling and undirected graphical models. They are widely used across structured prediction domains such as natural language processing, computer vision, and bioinformatics. The key idea in this class of models is to represent the joint distribution of a set of response variables *conditioned* on a set of covariates using a product of clique-wise compatibility functions. Given an underlying graph over the response variables, each of these compatibility functions depends on all the covariates, but only on a subset of response variables within any clique of the underlying graph. They are thus a *discriminative* counterpart of undirected graphical models, where we have covariates that provide information about the multivariate response, and the underlying graph structure encodes conditional independence assumptions among the responses conditioned on the covariates.

There is a key model specification question that arises, however, in any application of CRFs: how do we specify the clique-wise sufficient statistics, or compatibility functions (sometimes also called feature functions), that characterize the conditional graphical model between responses? In par-

ticular, how do we tune these to the particular types of variables being modeled? Traditionally, these questions have been addressed either by hand-crafted feature functions, or more generally by discretizing the multivariate response vectors into a set of indicator vectors and then letting the compatibility functions be linear combinations of the product of indicator functions [1]. This approach, however, may not be natural for continuous, skewed continuous or count-valued random variables. Recently, spurred in part by applications in bioinformatics, there has been much research on other sub-classes of CRFs. The Ising CRF which models binary responses, was studied by [2] and extended to higher-order interactions by [3]. Several versions and extensions of Gaussian-based CRFs have also been proposed [4, 5, 6, 7, 8]. These sub-classes of CRFs, however, are specific to Gaussian and binary variable types, and may not be appropriate for multivariate count data or skewed continuous data, for example, which are increasingly seen in big-data settings such as high-throughput genomic sequencing.

In this paper, we seek to (a) formulate a novel subclass of CRFs that have the flexibility to model responses of varied types, (b) address how to specify compatibility functions for such a family of CRFs, and (c) develop a tractable procedure with strong statistical guarantees for learning this class of CRFs from data. We first show that when node-conditional distributions of responses conditioned on other responses and covariates are specified by univariate exponential family distributions, there exists a consistent joint CRF distribution, that *necessarily has a specific form*: with terms that are tensorial products of functions over the responses, and functions over the covariates. This subclass of "exponential family" CRFs can be viewed as a conditional extension of the MRF framework of [9, 10]. As such, this broadens the class of off-the-shelf CRF models to encompass data that follows distributions other than the standard discrete, binary, or Gaussian instances. Given this new family of CRFs, we additionally show that if covariates also follow node-conditional univariate exponential family distributions, then the functions over features in turn are *precisely specified* by the exponential family sufficient statistics. Thus, our twin results definitively answer for the first time the key model specification question of specifying compatibility or feature functions for a broad family of CRF distributions. We then provide a unified $M$-estimation procedure, via penalized neighborhood estimation, to learn our family of CRFs from i.i.d. observations that simultaneously addresses all three sub-tasks of CRF learning: feature selection (where we select a subset of the covariates for any response variable), structure recovery (where we learn the graph structure among the response variables), and parameter learning (where we learn the parameters specifying the CRF distribution). We also present a single theorem that gives statistical guarantees saying that with high-probability, our $M$-estimator achieves each of these three sub-tasks. Our result can be viewed as an extension of neighborhood selection results for MRFs [11, 12, 13]. Overall, this paper provides a family of CRFs that generalizes many of the sub-classes in the existing literature and broadens the utility and applicability of CRFs to model many other types of multivariate responses.

## 2 Conditional Graphical Models via Exponential Families

Suppose we have a $p$-variate random response vector $Y = (Y_1, \ldots, Y_p)$, with each response variable $Y_s$ taking values in a set $\mathcal{Y}_s$. Suppose we also have a set of covariates $X = (X_1, \ldots, X_q)$ associated with this response vector $Y$. Suppose $G = (V, E)$ is an undirected graph over $p$ nodes corresponding to the $p$ response variables. Given the underlying graph $G$, and the set of cliques (fully-connected sub-graphs) $\mathcal{C}$ of the graph $G$, the corresponding conditional random field (CRF) is a set of distributions over the response conditioned on the covariates that satisfy Markov independence assumptions with respect to the graph $G$. Specifically, letting $\{\phi_c(Y_c, X)\}_{c \in \mathcal{C}}$ denote a set of clique-wise sufficient statistics, any strictly positive distribution of $Y$ conditioned on $X$ within the conditional random field family takes the form: $P(Y|X) \propto \exp\{\sum_{c \in \mathcal{C}} \phi_c(Y_c, X)\}$. With a pair-wise conditional random field distribution, the set of cliques consists of the set of nodes $V$ and the set of edges $E$, so that

$$P(Y|X) \propto \exp\left\{ \sum_{s \in V} \phi_s(Y_s, X) + \sum_{(s,t) \in E} \phi_{st}(Y_s, Y_t, X) \right\}.$$

A key model specification question is how to select the class of sufficient statistics, $\phi$. We have a considerable understanding of how to specify *univariate* distributions over various types of variables as well as on how to model their conditional response through regression. Consider the univariate exponential family class of distributions: $P(Z) = \exp(\theta B(Z) + C(Z) - D(\theta))$, with sufficient

statistics $B(Z)$, base measure $C(Z)$, and log-normalization constant $D(\theta)$. Such exponential family distributions include a wide variety of commonly used distributions such as Gaussian, Bernoulli, multinomial, Poisson, exponential, gamma, chi-squared, beta, any of which can be instantiated with particular choices of the functions $B(\cdot)$, and $C(\cdot)$. Such univariate exponential family distributions are thus used to model a wide variety of data types including skewed continuous data and count data. Additionally, through generalized linear models, they are used to model the response of various data types conditional on a set of covariates. Here, we seek to use our understanding of univariate exponential families and generalized linear models to specify a conditional graphical model distribution.

Consider the conditional extension of the construction in [14, 9, 10]. Suppose that the node-conditional distributions of response variables, $Y_s$, conditioned on the rest of the response variables, $Y_{V\setminus s}$, and the covariates, $X$, is given by an univariate exponential family:

$$P(Y_s|Y_{V\setminus s}, X) = \exp\{E_s(Y_{V\setminus s}, X)\, B_s(Y_s) + C_s(Y_s) - \bar{D}_s(Y_{V\setminus s}, X)\}. \tag{1}$$

Here, the functions $B_s(\cdot), C_s(\cdot)$ are specified by the choice of the exponential family, and the parameter $E_s(Y_{V\setminus s}, X)$ is an *arbitrary function* of the variables $Y_t$ in $N(s)$ and the covariates $X$; $N(s)$ is the set of neighbors of node $s$ according to an undirected graph $G = (V, E)$. Would these node-conditional distributions be consistent with a joint distribution? Would this joint distribution factor according a conditional random field given by graph $G$? And would there be restrictions on the form of the functions $E_s(Y_{V\setminus s}, X)$? The following theorem answers these questions. We note that it generalizes the MRF framework of [9, 10] in two ways: it allows for the presence of conditional covariates, and moreover allows for heterogeneous types and domains of distributions with the different choices of $B_s(\cdot)$ and $C_s(\cdot)$ at each individual node.

**Theorem 1.** *Consider a $p$-dimensional random vector $Y = (Y_1, Y_2, \ldots, Y_p)$ denoting the set of responses, and let $X = (X_1, \ldots, X_q)$ be a $q$-dimensional covariate vector. Consider the following two assertions: (a) the node-conditional distributions of each $P(Y_s|Y_{V\setminus s}, X)$ are specified by univariate exponential family distributions as detailed in (1); and (b) the joint multivariate conditional distribution $P(Y|X)$ factors according to the graph $G = (V, E)$ with clique-set $\mathcal{C}$, but with factors over response-variable-cliques of size at most $k$. These assertions on the conditional and joint distributions respectively are consistent if and only if the conditional distribution in (1) has the tensor-factorized form:*

$$P(Y_s|Y_{V\setminus s}, X; \theta) = \exp\Bigg\{ B_s(Y_s)\Big(\theta_s(X) + \sum_{t\in N(s)} \theta_{st}(X)\, B_t(Y_t) + \ldots$$

$$+ \sum_{t_2,\ldots,t_k\in N(s)} \theta_{s\,t_2\ldots t_k}(X) \prod_{j=2}^{k} B_{t_j}(Y_{t_j})\Big) + C_s(Y_s) - \bar{D}_s(Y_{V\setminus s})\Bigg\}, \tag{2}$$

*where $\theta_{s\cdot}(X) := \{\theta_s(X), \theta_{st}(X), \ldots, \theta_{s\,t_2\ldots t_k}(X)\}$ is a set of functions that depend only on the covariates $X$. Moreover, the corresponding joint conditional random field distribution has the form:*

$$P(Y|X; \theta) = \exp\Bigg\{ \sum_s \theta_s(X) B_s(Y_s) + \sum_{s\in V}\sum_{t\in N(s)} \theta_{st}(X)\, B_s(Y_s) B_t(Y_t)$$

$$+ \ldots + \sum_{(t_1,\ldots,t_k)\in\mathcal{C}} \theta_{t_1\ldots t_k}(X) \prod_{j=1}^{k} B_{t_j}(Y_{t_j}) + \sum_s C_s(Y_s) - A\big(\theta(X)\big)\Bigg\}, \tag{3}$$

*where $A\big(\theta(X)\big)$ is the log-normalization constant.*

Theorem 1 specifies the form of the function $E_s(Y_{V\setminus s}, X)$ defining the canonical parameter in the univariate exponential family distribution (1). This function is a *tensor factorization* of products of sufficient statistics of $Y_{V\setminus s}$, and "observation functions", $\theta(X)$, of the covariates $X$ alone. A key point to note is that the observation functions, $\theta(X)$, in the CRF distribution (3) should ensure that the density is normalizable, that is, $A\big(\theta(X)\big) < +\infty$. We also note that we can allow different exponential families for each of the node-conditional distributions of the response variables, meaning that the domains, $\mathcal{Y}_s$, or the sufficient statistics functions, $B_s(\cdot)$, can vary across the response variables $Y_s$. A common setting of these sufficient statistics functions however, for many popular distributions (Gaussian, Bernoulli, etc.), is a linear function, so that $B_s(Y_s) = Y_s$.

An important special case of the above result is when the joint CRF has response-variable-clique factors of size at most two. The node conditional distributions (2) would then have the form:

$$P(Y_s|Y_{V\setminus s}, X; \theta) \propto \exp\left\{ B_s(Y_s) \cdot \left( \theta_s(X) + \sum_{t \in N(s)} \theta_{st}(X) B_t(Y_t) \right) + C_s(Y_s) \right\},$$

while the joint distribution in (3) has the form:

$$P(Y|X; \theta) = \exp\left\{ \sum_{s \in V} \theta_s(X) B_s(Y_s) + \sum_{(s,t) \in E} \theta_{st}(X) B_s(Y_s) B_t(Y_t) + \sum_{s \in V} C_s(Y_s) - A\big(\theta(X)\big) \right\}, \quad (4)$$

with the log-partition function, $A\big(\theta(X)\big)$, given the covariates, $X$, defined as

$$A\big(\theta(X)\big) := \log \int_{\mathcal{Y}^p} \exp\left\{ \sum_{s \in V} \theta_s(X) B_s(Y_s) + \sum_{(s,t) \in E} \theta_{st}(X) B_s(Y_s) B_t(Y_t) + \sum_{s \in V} C_s(Y_s) \right\}. \quad (5)$$

Theorem 1 then addresses the model specification question of how to select the compatibility functions in CRFs for varied types of responses. Our framework permits arbitrary observation functions, $\theta(X)$, with the only stipulation that the log-partition function must be finite. (This only provides a restriction when the domain of the response variables is not finite). In the next section, we address the second model specification question of how to set the covariate functions.

## 2.1  Setting Covariate Functions

A candidate approach to specifying the observation functions, $\theta(X)$, in the CRF distribution above would be to make distributional assumptions on $X$. Since Theorem 1 specifies the conditional distribution $P(Y|X)$, specifying the *marginal distribution* $P(X)$ would allow us to specify the joint distribution $P(Y, X)$ without further restrictions on $P(Y|X)$ using the simple product rule: $P(X, Y) = P(Y|X) P(X)$. As an example, suppose that the covariates $X$ follow an MRF distribution with graph $G' = (V', E')$, and parameters $\vartheta$:

$$P(X) = \exp\left\{ \sum_{u \in V'} \vartheta_u \phi_u(X_u) + \sum_{(u,v) \in V' \times V'} \vartheta_{uv} \phi_{uv}(X_u, X_v) - A(\vartheta) \right\}.$$

Then, for any CRF distribution $P(Y|X)$ in (4), we have

$$P(X, Y) = \exp\left\{ \sum_u \vartheta_u \phi_u(X_u) + \sum_{(u,v)} \vartheta_{uv} \phi_{uv}(X_u, X_v) + \sum_s \theta_s(X) Y_s + \sum_{(s,t)} \theta_{st}(X) Y_s Y_t \right.$$
$$\left. + \sum_s C_s(Y_s) - A(\vartheta) - A\big(\theta(X)\big) \right\}.$$

The joint distribution, $P(X, Y)$, is valid provided $P(Y|X)$ and $P(X)$ are valid distributions. Thus, a distributional assumption on $P(X)$ does not restrict the set of covariate functions in any way.

On the other hand, specifying the conditional distribution, $P(X|Y)$, naturally entails restrictions on the form of $P(Y|X)$. Consider the case where the conditional distributions $P(X_u|X_{V'\setminus u}, Y)$ are also specified by univariate exponential families:

$$P(X_u|X_{V'\setminus u}, Y) = \exp\{ E_u(X_{V'\setminus u}, Y) B_u(X_u) + C_u(X_u) - \bar{D}_u(X_{V'\setminus u}, Y) \}, \quad (6)$$

where $E_u(X_{V'\setminus u}, Y)$ is an arbitrary function of the rest of the variables, and $B_u(\cdot), C_u(\cdot), \bar{D}_u(\cdot)$ are specified by the univariate exponential family. Under these additional distributional assumptions in (6), what form would the CRF distribution in Theorem 1 take? Specifically, what would be the form of the observation functions $\theta(X)$? The following theorem provides an answer to this question. (In the following, we use the shorthand $s_1^m$ to denote the sequence $(s_1, \ldots, s_m)$.)

**Theorem 2.** *Consider the following assertions: (a) the conditional CRF distribution of the responses $Y = (Y_1, \ldots, Y_p)$ given covariates $X = (X_1, \ldots, X_q)$ is given by the family (4); and (b) the conditional distributions of individual covariates given rest of the variables $P(X_u|X_{V'\setminus u}, Y)$ is given by an exponential family of the form in (6); and (c) the joint distribution $P(X, Y)$ belongs to a graphical model with graph $\bar{G} = (V \cup V', \bar{E})$, with clique-set $\mathcal{C}$, with factors of size at most $k$. These assertions are consistent if and only if the CRF distribution takes the form:*

$$P(Y|X) = \exp\left\{ \sum_{l=1}^{k} \sum_{\substack{t_1^r \in V, s_1^{l-r} \in V' \\ (t_1^r, s_1^{l-r}) \in \mathcal{C}}} \alpha_{t_1^r, s_1^{l-r}} \prod_{j=1}^{l-r} B_{s_j}(X_{s_j}) \prod_{j=1}^{r} B_{t_j}(Y_{t_j}) + \sum_{t \in V} C_t(Y_t) - A(\alpha, X) \right\}, \quad (7)$$

*so that the observation functions $\theta_{t_1,\ldots,t_r}(X)$ in the CRF distribution (4) are tensor products of univariate functions:* $\theta_{t_1,\ldots,t_r}(X) = \sum_{l=1}^{k} \sum_{\substack{s_1^{l-r} \in V' \\ (t_1^r, s_1^{l-r}) \in \mathcal{C}}} \alpha_{t_1^r, s_1^{l-r}} \prod_{j=1}^{l-r} B_{s_j}(X_{s_j}).$

Let us examine the consequences of this theorem for the pair-wise CRF distributions (4). Theorem 2 then entails that the observation functions, $\{\theta_s(X), \theta_{st}(X)\}$, have the following form when the distribution has factors of size at most two:

$$\theta_s(X) = \theta_s + \sum_{u \in V'} \theta_{su} B_u(X_u), \quad \theta_{st}(X) = \theta_{st}, \quad (8)$$

for some *constant* parameters $\theta_s$, $\theta_{su}$ and $\theta_{st}$. Similarly, if the joint distribution has factors of size at most three, we have:

$$\theta_s(X) = \theta_s + \sum_{u \in V'} \theta_{su} B_u(X_u) + \sum_{(u,v) \in V' \times V'} \theta_{suv} B_u(X_u) B_v(X_v),$$

$$\theta_{st}(X) = \theta_{st} + \sum_{u \in V'} \theta_{stu} B_u(X_u). \quad (9)$$

**(Remark 1)** While we have derived the covariate functions in Theorem 2 by assuming a distributional form on $X$, using the resulting covariate functions do not necessarily impose distributional assumptions on $X$. This is similar to "generative-discriminative" pairs of models [15]: a "generative" Naive Bayes distribution for $P(X|Y)$ corresponds to a "discriminative" logistic regression model for $P(Y|X)$, but the converse need not hold. We can thus leverage the parametric CRF distributional form in Theorem 2 without necessarily imposing stringent distributional assumptions on $X$.

**(Remark 2)** Consider the form of the covariate functions given by (8) compared to (9). What does sparsity in the parameters entail in terms of conditional independence assumptions? $\theta_{st} = 0$ in (8) entails that $Y_s$ is conditionally independent of $Y_t$ given the other responses and all the covariates. Thus, the parametrization in (8) corresponds to pair-wise conditional independence assumptions between the responses (structure learning) and between the responses and covariates (feature selection). In contrast, (9) lets the edges weights between the responses, $\theta_{st}(X)$ vary as a linear combination of the covariates. Letting $\theta_{stu} = 0$ entails the lack of a third-order interaction between the pair of responses $Y_s$ and $Y_t$ and the covariate $X_u$, conditioned on all other responses and covariates.

**(Remark 3)** Our general subclasses of CRFs specified by Theorems 1 and 2 encompass many existing CRF families as special cases, in addition to providing many novel forms of CRFs.

- The Gaussian CRF presented in [7] as well as the reparameterization in [8] can be viewed as an instance of our framework by substituting in Gaussian sufficient statistics in (8): here the Gaussian mean of the CRF depends on the covariates, but not the covariance. We can correspondingly derive a *novel Gaussian CRF* formulation from (9), where the Gaussian covariance of $Y|X$ would also depend on $X$.

- By using the Bernoulli distribution as the node-conditional distribution, we can derive the Ising CRF, recently studied in [2] with an application to studying tumor suppressor genes.

- Several novel forms of CRFs can be derived by specifying node-conditional distributions as Poisson or exponential, for example. With certain distributions, such as the multivariate Poisson for example, we would have to enforce constraints on the parameters to ensure normalizability of the distribution. For the Poisson CRF distribution, it can be verified that for the log-partition function to be finite, $A(\theta_{st}(X)) < \infty$, the observation functions are constrained to be non-positive, $\theta_{st}(X) \leq 0$. Such restrictions are typically needed for cases where the variables have infinite domains.

# 3   Graphical Model Structure Learning

We now address the task of learning a CRF distribution from our general family given i.i.d. observations of the multivariate response vector and covariates. Structure recovery and estimation for CRFs has not attracted as much attention as that for MRFs. Schmidt et al. [16], Torralba et al. [17] empirically study greedy methods and block $\ell_1$ regularized pseudo-likelihood respectively to learn the discrete CRF graph structure. Bradley and Guestrin [18], Shahaf et al. [19] provide guarantees on structure recovery for low tree-width discrete CRFs using graph cuts, and a maximum weight spanning tree based method respectively. Cai et al. [4], Liu et al. [6] provide structure recovery guarantees for their two-stage procedure for recovering (a reparameterization of) a conditional Gaussian based CRF; and the semi-parameteric partition based Gaussian CRF respectively. Here, we provide a single theorem that provides structure recovery guarantees for any CRF from our class of exponential family CRFs, which encompasses not only Ising, and Gaussian based CRFs, but all other instances within our class, such as Poisson CRFs, exponential CRFs, and so on.

We are given $n$ i.i.d. samples $\mathcal{Z} := \{X^{(i)}, Y^{(i)}\}_{i=1}^n$ from a pair-wise CRF distribution, of the form specified by Theorems 1 and 2 with covariate functions as given in (8):

$$P(Y|X;\theta^*) \propto \exp\left\{\sum_{s\in V}\left(\theta_s^* + \sum_{u\in N'(s)}\theta_{su}^* B_u(X_u)\right)B_s(Y_s) + \sum_{(s,t)\in E}\theta_{st}^* B_s(Y_s)B_t(Y_t) + \sum_s C(Y_s)\right\}, \quad (10)$$

with unknown parameters, $\theta^*$. The task of CRF parameter learning corresponds to estimating the parameters $\theta^*$, structure learning corresponds to recovering the edge-set $E$, and feature selection corresponds to recovering the neighborhoods $N'(s)$ in (10). Note that the log-partition function $A(\theta^*)$ is intractable to compute in general (other than special cases such as Gaussian CRFs). Accordingly, we adopt the node-based neighborhood estimation approach of [12, 13, 9, 10]. Given the joint distribution in (10), the node-wise conditional distribution of $Y_s$ given the rest of the nodes and covariates, is given by $P(Y_s|Y_{V\setminus s}, X; \theta^*) = \exp\{\eta \cdot B_s(Y_s) + C_s(Y_s) - D_s(\eta)\}$ which is a univariate exponential family, with parameter $\eta = \theta_s^* + \sum_{u\in V'}\theta_{su}^* B_u(X_u) + \sum_{t\in V\setminus s}\theta_{st}^* B_t(Y_t)$, as discussed in the previous section. The corresponding negative log-conditional-likelihood can be written as $\ell(\theta; \mathcal{Z}) := -\frac{1}{n}\log\prod_{i=1}^n P(Y_s^{(i)}|Y_{V\setminus s}^{(i)}, X^{(i)}; \theta)$.

For each node $s$, we have three components of the parameter set, $\boldsymbol{\theta} := (\theta_s, \boldsymbol{\theta}^x, \boldsymbol{\theta}^y)$: a scalar $\theta_s$, a length $q$ vector $\boldsymbol{\theta}^x := \cup_{u\in V'}\theta_{su}$, and a length $p-1$ vector $\boldsymbol{\theta}^y := \cup_{t\in V\setminus s}\theta_{st}$. Then, given samples $\mathcal{Z}$, these parameters can be selected by the following $\ell_1$ regularized $M$-estimator:

$$\min_{\boldsymbol{\theta}\in\mathbb{R}^{1+(p-1)+q}} \ell(\boldsymbol{\theta}) + \lambda_{x,n}\|\boldsymbol{\theta}^x\|_1 + \lambda_{y,n}\|\boldsymbol{\theta}^y\|_1, \quad (11)$$

where $\lambda_{x,n}$, $\lambda_{y,n}$ are the regularization constants. Note that $\lambda_{x,n}$ and $\lambda_{y,n}$ do not need to be the same as $\lambda_{y,n}$ determines the degree of sparsity between $Y_s$ and $Y_{V\setminus s}$, and similarly $\lambda_{x,n}$ does the degree of sparsity between $Y_s$ and covariates $X$. Given this $M$-estimator, we can recover the response-variable-neighborhood of response $Y_s$ as $N(s) = \{t \in V\setminus s \mid \theta_{st}^y \neq 0\}$, and the feature-neighborhood of the response $Y_s$ as $N'(s) = \{t \in V' \mid \theta_{su}^x \neq 0\}$.

Armed with this machinery, we can provide the statistical guarantees on successful learning of all three sub-tasks of CRFs:

**Theorem 3.** *Consider a CRF distribution as specified in* (10). *Suppose that the regularization parameters in* (11) *are chosen such that*

$$\lambda_{x,n} \geq M_1\sqrt{\frac{\log q}{n}}, \quad \lambda_{y,n} \geq M_1\sqrt{\frac{\log p}{n}} \quad and \quad \max\{\lambda_{x,n}, \lambda_{y,n}\} \leq M_2,$$

*where $M_1$ and $M_2$ are some constants depending on the node conditional distribution in the form of exponential family. Further suppose that $\min_{t\in N(s)}|\theta_{st}^*| \geq \frac{10}{\rho_{\min}}\max\{\sqrt{d_x}\lambda_{x,n}, \sqrt{d_y}\lambda_{y,n}\}$ where $\rho_{\min}$ is the minimum eigenvalue of the Hessian of the loss function at $\boldsymbol{\theta}^{x*}, \boldsymbol{\theta}^{y*}$, and $d_x$, $d_y$ are the number of nonzero elements in $\boldsymbol{\theta}^{x*}$ and $\boldsymbol{\theta}^{y*}$, respectively. Then, for some positive constants $L$, $c_1$, $c_2$, and $c_3$, if $n \geq L(d_x + d_y)^2(\log p + \log q)(\max\{\log n, \log(p+q)\})^2$, then with probability at least $1 - c_1\max\{n, p+q\}^{-2} - \exp(-c_2 n) - \exp(-c_3 n)$, the following statements hold.*

**(a)** *(Parameter Error) For each node $s \in V$, the solution $\widehat{\boldsymbol{\theta}}$ of the M-estimation problem in* (11) *is unique with parameter error bound*

$$\|\widehat{\boldsymbol{\theta}^x} - \boldsymbol{\theta}^{x*}\|_2 + \|\widehat{\boldsymbol{\theta}^y} - \boldsymbol{\theta}^{y*}\|_2 \leq \frac{5}{\rho_{\min}}\max\{\sqrt{d_x}\lambda_{x,n}, \sqrt{d_y}\lambda_{y,n}\}$$

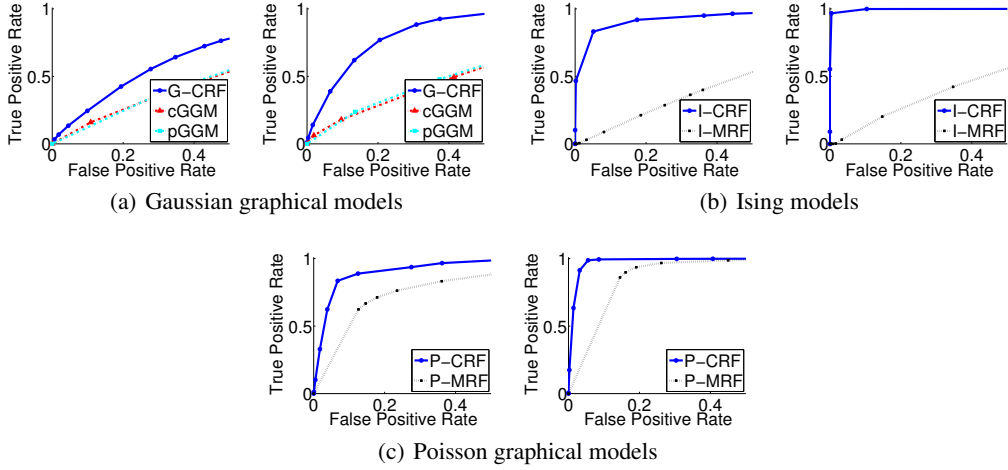

Figure 1: (a) ROC curves averaged over 50 simulations from a Gaussian CRF with $p = 50$ responses, $q = 50$ covariates, and (left) $n = 100$ and (right) $n = 250$ samples. Our method (G-CRF) is compared to that of [7] (cGGM) and [8] (pGGM). (b) ROC curves for simulations from an Ising CRF with $p = 100$ responses, $q = 10$ covariates, and (left) $n = 50$ and (right) $n = 150$ samples. Our method (I-CRF) is compared to the unconditional Ising MRF (I-MRF). (c) ROC curves for simulations from a Poisson CRF with $p = 100$ responses, $q = 10$ covariates, and (left) $n = 50$ and (right) $n = 150$ samples. Our method (P-CRF) is compared to the Poisson MRF (P-MRF).

**(b)** *(Structure Recovery) The M-estimate recovers the response-feature neighborhoods exactly, so that* $\widehat{N'}(s) = N'(s)$, *for all* $s \in V$.

**(c)** *(Feature Selection) The M-estimate recovers the true response neighborhoods exactly, so that* $\widehat{N}(s) = N(s)$, *for all* $s \in V$.

The proof requires modifying that of Theorem 1 in [9, 10] to allow for two different regularization parameters, $\lambda_{x,n}$ and $\lambda_{y,n}$, and for two distinct sets of random variables (responses and covariates). This introduces subtleties related to interactions in the analyses. Extending our statistical analysis in Theorem 3 for pair-wise CRFs to general CRF distributions (3) as well as general covariate functions, such as in (9), are omitted for space reasons and left for future work.

## 4 Experiments

**Simulation Studies.** In order to evaluate the generality of our framework, we simulate data from three different instances of our model: those given by Gaussian, Bernoulli (Ising), and Poisson node-conditional distributions. We assume the true conditional distribution, $P(Y|X)$, follows (7) with the parameters: $\theta_s(X) = \theta_s + \sum_{u \in V'} \theta_{su} X_u$, $\theta_{st}(X) = \theta_{st} + \sum_{u \in V'} \theta_{stu} X_u$ for some *constant* parameters $\theta_s$, $\theta_{su}$, $\theta_{st}$ and $\theta_{stu}$. In other words, we permit both the mean, $\theta_s(X)$ and the covariance or edge-weights, $\theta_{st}(X)$, to depend on the covariates.

For the Gaussian CRFs, our goal is to infer the precision (or inverse covariance) matrix. We first generate covariates as $X \sim U[-0.05, 0.05]$. Given $X$, the precision matrix of $Y$, $\Theta(X)$, is generated as follows. All the diagonal elements are set to 1. For each node $s$, 4 nearest neighbors in the $\sqrt{p} \times \sqrt{p}$ lattice structure are selected, and $\theta_{st} = 0$ for non-neighboring nodes. For a given edge structure, the strength is now a function of covariates, $X$, by letting $\theta_{st}(X) = c + \langle \omega_{st}, X \rangle$ where $c$ is a constant bias term and $\omega_{st}$ is target vector of length $q$. Data of size $p = 50$ responses and $q = 50$ covariates was generated for $n = 100$ and $n = 250$ samples. Figure 1(a) reports the receiver-operator curves (ROC) averaged over 50 trials for three different methods: the model of [7] (denoted as cGGM), the model of [8] (denoted as pGGM), and our method (denoted as G-CRF). Results show that our method outperforms competing methods as their edge-weights are restricted to be constants, while our method allows them to linearly depend on the covariates. Data was similarly generated using a 4 nearest neighbor lattice structure for Ising and Poisson CRFs with $p = 100$ responses,

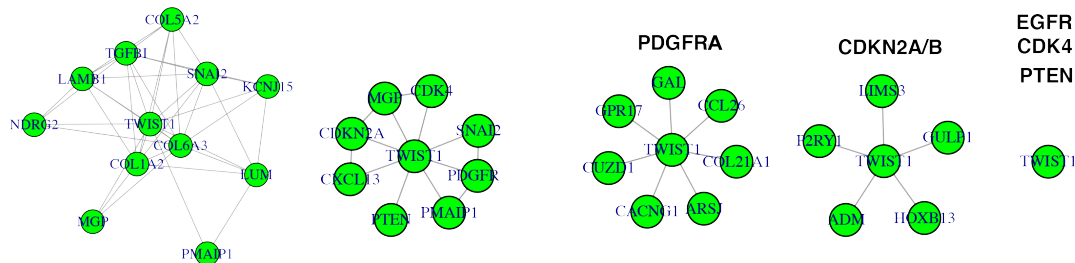

Figure 2: From left to right: Gaussian MRF, mean-specified Gaussian CRF, and the set corresponding to the covariance-specified Gaussian CRF. The latter shows the third-order interactions between gene-pairs and each of the five common aberration covariates (EGFR, PTEN, CDKN2A, PDGFRA, and CDK4). The models were learned from gene expression array data of Glioblastoma samples, and the plots display the response neighborhoods of gene TWIST1.

$q = 10$ covariates, and $n = 50$ or $n = 150$ samples. Figure 1(b) and Figure 1(c) report the ROC curves averaged over 50 trials for the Ising and Poisson CRFs respectively. The performance of our method is compared to that of the unconditional Ising and Poisson MRFs of [9, 10].

**Real Data Example: Genetic Networks of Glioblastoma.** We demonstrate the performance of our CRF models by learning genetic networks of Glioblastoma conditioned on common copy number aberrations. Level III gene expression data measured by Aglient arrays for $n = 465$ Glioblastoma tumor samples as well as copy number variation measured by CGH-arrays were downloaded from the Cancer Genome Atlas data portal [20]. The data was processed according to standard techniques, and we only consider genes from the C2 Pathway Database. The five most common copy number aberrations across all subjects were taken as covariates. We fit our Gaussian "mean-specified" CRFs (with covariate functions given in (8)) and Gaussian "covariance-specified" CRFs (with covariate functions given in (9)) by penalized neighborhood estimation to learn the graph structure of gene expression responses, $p = 876$, conditional on $q = 5$ aberrations: EGFR, PTEN, CDKN2A, PDGFRA, and CDK4. Stability selection [21] was used to determine the sparsity of the network.

Due to space limitations, the entire network structures are not shown. Instead, we show the results of the mean- and covariance-specified Gaussian CRFs and that of the Gaussian graphical model (GGM) for one particularly important gene neighborhood: TWIST1 is a transcription factor for epithelial to mesenchymal transition [22] and has been shown to promote tumor invasion in multiple cancers including Glioblastoma [23]. The neighborhoods of TWIST1 learned by GGMs and mean-specified CRFs share many of the known interactors of TWIST1, such as SNAI2, MGP, and PMAIP1 [24]. The mean-specified CRF is more sparse as conditioning on copy number aberrations may explain many of the conditional dependencies with TWIST1 that are captured by GGMs, demonstrating the utility of conditional modeling via CRFs. For the covariance-specified Gaussian CRF, we plot the neighborhood given by $\theta_{stu}$ in (9) for the five values of $u$ corresponding to each aberration. The results of this network denote third-order effects between gene-pairs and aberrations, and are thus even more sparse with no neighbors for the interactions between TWIST1 and PTEN, CDK4, and EGFR. TWIST1 has different interactions between PDGFRA and CDKN2A, which have high frequency for proneual subtypes of Glioblastoma tumors. Thus, our covariance-specified CRF network may indicate that these two aberrations are the most salient in interacting with pairs of genes that include the gene TWIST1. Overall, our analysis has demonstrated the applied advantages of our CRF models; namely, one can study the network structure between responses conditional on covariates and/or between pairs of responses that interact with particular covariates.

## Acknowledgments

The authors acknowledge support from the following sources: ARO via W911NF-12-1-0390 and NSF via IIS-1149803 and DMS-1264033 to E.Y. and P.R; Ken Kennedy Institute for Information Technology at Rice to G.A. and Z.L.; NSF DMS-1264058 and DMS-1209017 to G.A.; and NSF DMS-1263932 to Z.L..

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
