[Supplementary Material]

# Appendix

## A  Proof of Theorem 1

This theorem can be understood as the extension of Proposition 2 in [9]. We follow the proof policy of that paper: Define $Q(\mathbf{y}|x)$ as

$$Q(\mathbf{y}|x) := \log(P(\mathbf{y}|x)/P(\mathbf{0}|x)),$$

for any $\mathbf{y} = (y_1, \ldots, y_p) \in \mathcal{Y}^p$ given $x$ where $\mathbf{0}$ indicates a zero vector (The number of zeros vary appropriately in the context below). For any $\mathbf{y}$, also denote $\bar{\mathbf{y}}_s := (y_1, \ldots, y_{s-1}, 0, y_{s+1}, \ldots, y_p)$.

Now, consider the following general form for $Q(\mathbf{y}|x)$:

$$Q(\mathbf{y}|x) = \sum_{t_1 \in V} y_{t_1} G_{t_1}(y_{t_1}, x) + \ldots + \tag{12}$$

$$\sum_{t_1,\ldots,t_k \in V} y_{t_1} \ldots y_{t_k} G_{t_1,\ldots,t_k}(y_{t_1}, \ldots, y_{t_k}, x),$$

since the joint distribution on $Y$ given $X$ has factors of size $k$ at most. It can then be seen that

$$\exp(Q(\mathbf{y}|x) - Q(\bar{\mathbf{y}}_s|x)) = P(\mathbf{y}|x)/P(\bar{\mathbf{y}}_s|x)$$
$$= \frac{P(y_s|y_1, \ldots, y_{s-1}, y_{s+1}, \ldots, y_p, x)}{P(0|y_1, \ldots, y_{s-1}, y_{s+1}, \ldots, y_p, x)}, \tag{13}$$

where the first equality follows from the definition of $Q$, and the second equality follows from some algebra. Now, consider simplifications of both sides of (13). Given the form of $Q(\mathbf{y}|x)$ in (12), we have

$$Q(\mathbf{y}|x) - Q(\bar{\mathbf{y}}_1|x) = \tag{14}$$

$$y_1 \Bigg( G_1(y_1, x) + \sum_{t=2}^{p} y_t G_{1t}(y_1, y_t, x) + \ldots +$$

$$\sum_{t_2,\ldots,t_k \in \{2,\ldots,p\}} y_{t_2} \ldots y_{t_k} G_{1,t_2,\ldots,t_k}(y_1, \ldots, y_{t_k}, x) \Bigg).$$

Also, given the exponential family form of the node-conditional distribution specified in the theorem,

$$\log \frac{P(y_i|y_1, \ldots, y_{s-1}, y_{s+1}, \ldots, y_p, x)}{P(0|y_1, \ldots, y_{s-1}, y_{s+1}, \ldots, y_p, x)} = \tag{15}$$
$$E_s(y_{V \setminus s}, x)(B_s(y_s) - B_s(0)) + (C_s(y_s) - C_s(0)).$$

Setting $y_t = 0$ for all $t \neq s$ in (13), and using the expressions for the left and right hand sides in (14) and (15), we obtain,

$$y_s G_s(y_s, x)$$
$$= E_s(\mathbf{0}, x)(B_s(y_s) - B_s(0)) + (C_s(y_s) - C_s(0)).$$

Setting $y_r = 0$ for all $r \notin \{s, t\}$,

$$y_s G_s(y_s, x) + y_s y_t G_{st}(y_s, y_t, x)$$
$$= E_s(\mathbf{0}, y_t, \mathbf{0}, x)(B_s(y_s) - B_s(0)) + (C_s(y_s) - C_s(0)).$$

Combining these two equations yields

$$y_s y_t G_{st}(y_s, y_t, x)$$
$$= \big(E_s(\mathbf{0}, y_t, \mathbf{0}, x) - E_s(\mathbf{0}, x)\big)(B_s(y_s) - B_s(0)). \tag{16}$$

Similarly, from the same reasoning for node $t$, we have

$$y_t G_t(y_t, x) + y_s y_t G_{st}(y_s, y_t, x)$$
$$= E_t(\mathbf{0}, y_s, \mathbf{0}, x)(B_t(y_t) - B_t(0)) + (C_t(y_t) - C_t(0)),$$

and at the same time,

$$y_s y_t G_{st}(y_s, y_t, x)$$
$$= \big(E_t(\mathbf{0}, y_s, \mathbf{0}, x) - E_t(\mathbf{0}, x)\big)(B_t(y_t) - B_t(0)). \tag{17}$$

Therefore, from (16) and (17), we obtain

$$E_t(\mathbf{0}, y_s, \mathbf{0}, x) - E_t(\mathbf{0}, x)$$
$$= \frac{E_s(\mathbf{0}, y_t, \mathbf{0}, x) - E_s(\mathbf{0}, x)}{B_t(y_t) - B_t(0)}(B_s(y_s) - B_s(0)). \tag{18}$$

Since (18) should hold for all possible combinations of $y_s$, $y_t$ and $x$, for any fixed $y_t \neq 0$,

$$E_t(\mathbf{0}, y_s, \mathbf{0}, x) - E_t(\mathbf{0}, x)$$
$$= \theta_{st}(x)(B_s(y_s) - B_s(0)) \tag{19}$$

where $\theta_{st}(\cdot)$ is a function on $x$. Plugging (19) back into (17),

$$y_s y_t G_{st}(y_s, y_t, x)$$
$$= \theta_{st}(x)(B_s(y_s) - B_s(0))(B_t(y_t) - B_t(0)).$$

More generally, by considering non-zero triplets, and setting $y_r = 0$ for all $r \notin \{s, t, u\}$, we obtain,

$$y_s G_s(y_s, x) + y_s y_t G_{st}(y_s, y_t, x)$$
$$+ y_s y_u G_{su}(y_s, y_u, x) + y_s y_t y_u G_{stu}(y_s, y_t, y_u, x)$$
$$= E_s(\mathbf{0}, y_t, \mathbf{0}, y_u, \mathbf{0}, x)(B_s(y_s) - B_s(0))$$
$$+ (C_s(y_s) - C_s(0)),$$

so that by a similar reasoning we can obtain

$$y_s y_t y_u G_{stu}(y_s, y_t, y_u, x) =$$
$$\theta_{stu}(x)(B_s(y_s) - B_s(0))(B_t(y_t) - B_t(0))(B_u(y_u) - B_u(0)).$$

More generally, we can show that

$$y_{t_1} \ldots y_{t_k} G_{t_1, \ldots, t_k}(y_{t_1}, \ldots, y_{t_k}, x) =$$
$$\theta_{t_1, \ldots, t_k}(x)(B_{t_1}(y_{t_1}) - B_{t_1}(0)) \ldots (B_{t_k}(y_{t_k}) - B_{t_k}(0)).$$

Thus, the $k$-th order factors in the joint distribution as specified in (12) are tensor products of $(B_s(y_s) - B_s(0))$, thus proving the statement of the theorem.

## B    Proof of Theorem 3

### B.1    Conditions

A key quantity in the analysis is the Fisher Information matrix, $Q^* = \nabla^2 \ell(\boldsymbol{\theta}^*; \mathcal{Z})$, the Hessian of the node-conditional log-likelihood where the reference node $s$ should be understood implicitly. We use $S = \{(s, t) : t \in N(s)\}$ to denote the true neighborhood of node $s$, and $S^c$ to denote its complement. Similarly, we also use $T$ to denote non-zero element of $\boldsymbol{\theta}^x$, and $T^c$ for its complement. $Q_{SS}^*$ indicates $d_y \times d_y$ sub-matrix indexed by $S$ where $d_y$ is the maximum node degree. $Q_{TT}^*$ can be defined in a similar way, and so on. Our conditions mirror those in [10]:

**Condition 1** (Dependency condition). There exists a constant $\rho_{\min} > 0$ such that $\min\{\lambda_{\min}(Q_{SS}^*), \lambda_{\min}(Q_{TT}^*)\} \geq \rho_{\min}$ so that the sub-matrix of Fisher Information matrix corresponding to true neighborhood has bounded eigenvalues. Moreover, there exists a constant $\rho_{\max} < \infty$ such that $\lambda_{\max}(\widehat{\mathbb{E}}[[Y_{V \setminus s}; X][Y_{\setminus s}; X]^T]) \leq \rho_{\max}$.

These condition can be understood as ensuring that variables do not become overly dependent. We will also need an incoherence or irrepresentable condition on the Fisher information matrix as in [13].

**Condition 2** (Incoherence condition). There exists a constant $\alpha > 0$, such that $\max\big\{\max_{t \in S^c} \|Q_{tS}^*(Q_{SS}^*)^{-1}\|_1, \max_{v \in T^c} \|Q_{vT}^*(Q_{TT}^*)^{-1}\|_1\big\} \leq 1 - \alpha$.

This condition, standard in high-dimensional analyses, can be understood as ensuring that irrelevant variables do not exert an overly strong effect on the true neighboring variables.

For notational simplicity, let $Y'$ be the random vector including all random variables $Y$ as well as covariates $X$, and $G'' = (V'', E'')$ be the graph corresponding to the combined variables $X$ and $Y$. By Theorem 1 and the the node-conditional distributions specified in (10), the joint distribution $P(X, Y)$ and the node-conditional distributions should have the form:

$$P(Y'; \theta) = \exp \left\{ \sum_{s \in V''} \theta_s B_s(Y'_s) + \sum_{(s,t) \in E''} \theta_{st} B_s(Y'_s) B_t(Y'_t) + \sum_{s \in V''} C_s(Y'_s) - A(\theta) \right\}, \quad (20)$$

$$P(Y'_s | Y'_{V'' \backslash s}; \theta) = \exp \left\{ B_s(Y'_s) \cdot \eta + C_s(Y'_s) - D_s(\eta) \right\} \quad (21)$$

where $\eta = \theta_s + \sum_{t \in V'' \backslash s} \theta_{st} B_t(Y'_t)$.

The following two conditions are on the log-partitions of (20) and (21):

**Condition 3.** The log-partition function $A(\cdot)$ of the joint distribution of $P(X, Y)$ (20) satisfies: For all $s \in V \cup V'$, (i) there exist constants $\kappa_m$, $\kappa_v$ such that the first and the second moment satisfy $\mathbb{E}[Y'_s] \leq \kappa_m$ and $\mathbb{E}[Y'^2_s] \leq \kappa_v$, respectively. Additionally, we have a constant $\kappa_h$ for which $\max_{u:|u|\leq 1} \frac{\partial^2 A(\theta)}{\partial \theta_s^2} (\{\theta_s^* + u, \ \theta^*\}) \leq \kappa_h$, and (ii) for scalar variable $\eta$, we define a function which is slightly different from (5):

$$\bar{A}_s(\eta; \theta) := \log \int_{\mathcal{Y}'^p} \exp \left\{ \eta B_s(Y'_s)^2 + \sum_{s \in V''} \theta_s B_s(Y'_s) + \sum_{(s,t) \in E''} \theta_{st} B_s(Y'_s) B_t(Y'_t) + \sum_{s \in V''} C_s(Y'_s) \right\},$$
$$(22)$$

where $\nu$ is an underlying measure with respect to which the density is taken. Then, there exists a constant $\kappa_h$ such that $\max_{u:|u|\leq 1} \frac{\partial^2 \bar{A}_s(\eta; \theta^*)}{\partial \eta^2}(u) \leq \kappa_h$.

**Condition 4.** For all $s \in V$, the log-partition function $D(\cdot)$ of the node-wise conditional distribution (21) satisfies: there exist functions $\kappa_1(n, p)$ and $\kappa_2(n, p)$ (that depend on the exponential family) such that, for all feasible pairs of $\theta$ and $X$, $|D''(a)| \leq \kappa_1(n, p)$ where $a \in [b, b + 4\kappa_2(n, p) \max\{\log n, \log p\}]$ for $b := \theta_s + \langle \theta_{\backslash s}, X_{V'' \backslash s} \rangle$. Additionally, $|D'''(b)| \leq \kappa_3(n, p)$ for all feasible pairs of $\theta$ and $X$. Note that $\kappa_1(n, p), \kappa_2(n, p)$ and $\kappa_3(n, p)$ are functions that might be dependent on $n$ and $p$, which affect our main theorem below.

Conditions 3 and 4 are the key technical components enabling us to generalize the analyses in [11, 12, 13] to the general GLM case.

Armed with the conditions above, we can show that the random vectors $Y$ given $X$ following the conditional graphical model distribution in (10) are suitably well-behaved (the proof can be trivially extended from [10]):

**Proposition 1.** *Suppose $Y$ is a random vector with the distribution specified in (10). Further, we assume that the node-conditional distribution of $X_u$ has the exponential family form (6). Then, for $\delta \leq \min\{2\kappa_v/3, \kappa_h + \kappa_v\}$, and some constant $c > 0$,*

$$P \left( \frac{1}{n} \sum_{i=1}^n B_s \left( Y_s^{(i)} \right)^2 \geq \delta \right) \leq \exp \left( -c \, n \, \delta^2 \right), \quad P \left( \frac{1}{n} \sum_{i=1}^n B_u \left( X_u^{(i)} \right)^2 \geq \delta \right) \leq \exp \left( -c \, n \, \delta^2 \right).$$

*Furthermore, For any positive constant $\delta$, and some constant $c > 0$,*

$$P \Big( |B_s(Y_s)| \geq \delta \log \eta \Big) \leq c\eta^{-\delta}, \quad and \quad P \Big( |B_u(X_u)| \geq \delta \log \eta \Big) \leq c\eta^{-\delta}.$$

This proposition plays a key role in the proof of sparsistency result below.

## B.2 Proof of Theorem 3

Since two regularizers in the optimization problem (11) separately concern two distinct sets of parameters, the subgradient optimality condition from the convex objective can be written as

$$\nabla \ell(\widehat{\boldsymbol{\theta}}; \mathcal{Z}) + \begin{bmatrix} 0 \\ \lambda_{x,n} \widehat{Z}^x \\ \lambda_{y,n} \widehat{Z}^y \end{bmatrix} = 0, \tag{23}$$

where $\widehat{Z}^x$ is a subgradient vector corresponding to the parameter $\boldsymbol{\theta}^x$; if $\widehat{\theta}_{si} \neq 0$, then the corresponding element in $\widehat{Z}^x$ has $sign(\widehat{\theta}_{si})$, and its absolute value is smaller than 1 otherwise. $\widehat{Z}^y$ is defined in a similar way. In the high-dimensional regime with $p, q \gg n$, the objective function is not necessarily strictly convex, as a result, it might be the case that there are multiple optimal solutions satisfying (23). Nonetheless, we can complete the proof simply by using the *primal-dual witness* techniques used in the several past works [13, 25]; We only need to show the strict dual feasibility holds with high probability, for the optimal parameters solving the optimization problem with the knowledge of *unknown* support set.

In order to show the dual feasibility holds, i.e., $\|\widehat{Z}^x\|_\infty < 1$ and $\|\widehat{Z}^y\|_\infty < 1$ with high probability, we rewrite a subgradient condition (23) into a form easier to analyze:

$$\nabla^2 \ell(\boldsymbol{\theta}^*; \mathcal{Z})(\widehat{\boldsymbol{\theta}} - \boldsymbol{\theta}^*) + \begin{bmatrix} 0 \\ \lambda_{x,n} \widehat{Z}^x \\ \lambda_{y,n} \widehat{Z}^y \end{bmatrix} = \begin{bmatrix} W_1^n \\ W_x^n \\ W_y^n \end{bmatrix} + \begin{bmatrix} R_1^n \\ R_x^n \\ R_y^n \end{bmatrix}, \tag{24}$$

where $W^n$ represented as the vector form in the right-hand side is defined as $-\nabla \ell(\widehat{\boldsymbol{\theta}}; \mathcal{Z})$, and similarly $R^n$ is the remainder after the coordinate-wise application of the mean value theorems; $R_j^n = [\nabla^2 \ell(\boldsymbol{\theta}^*; \mathcal{Z}) - \nabla^2 \ell(\bar{\boldsymbol{\theta}}^{(j)}; \mathcal{Z})]_j^T (\widehat{\boldsymbol{\theta}} - \boldsymbol{\theta}^*)$, for some $\bar{\boldsymbol{\theta}}^{(j)}$ on the line between $\widehat{\boldsymbol{\theta}}$ and $\boldsymbol{\theta}^*$, and with $[\cdot]_j^T$ being the $j$-th row of a matrix.

In the sequel, we provide three lemmas that control the right-hand side of (24):

**Lemma 1.** *Suppose that we set $\lambda_{x,n}$ and $\lambda_{y,n}$ to satisfy:*

$$\lambda_{x,n} \geq \frac{8(2-\alpha)}{\alpha} \sqrt{\kappa_1(n,p)\kappa_4} \sqrt{\frac{\log q}{n}}, \quad \lambda_{y,n} \geq \frac{8(2-\alpha)}{\alpha} \sqrt{\kappa_1(n,p)\kappa_4} \sqrt{\frac{\log p}{n}} \text{ and}$$

$$\max\{\lambda_{x,n}, \lambda_{y,n}\} \leq \frac{4(2-\alpha)}{\alpha} \kappa_1(n,p)\kappa_2(n,p)\kappa_4,$$

*for some constant $\kappa_4 \leq \min\{2\kappa_v/3, 2\kappa_h + \kappa_v\}$. Suppose also that $n \geq \frac{8\kappa_h^2}{\kappa_4^2}(\log p + \log q)$. Then, given the mutual incoherence parameter $\alpha \in (0,1]$, and $p' := \max\{n, p+q\}$,*

$$P\left(\frac{2-\alpha}{\lambda_{x,n}} \|W_x^n\|_\infty \leq \frac{\alpha}{4}, \frac{2-\alpha}{\lambda_{y,n}} \|W_y^n\|_\infty \leq \frac{\alpha}{4}\right) \geq 1 - c_1 p'^{-2} - \exp(-c_2 n) - \exp(-c_3 n). \tag{25}$$

**Lemma 2.** *Suppose that $\sqrt{d_x + d_y} \max\left\{\sqrt{d_x}\lambda_{x,n}, \sqrt{d_y}\lambda_{y,n}\right\} \leq \frac{\rho_{\min}^2}{72\rho_{\max}\kappa_3(n,p)\log p'}$ and $\|W^n\|_\infty \leq \frac{\lambda_n}{4}$. Then, we have*

$$P\left(\|\widehat{\boldsymbol{\theta}}_S - \boldsymbol{\theta}_S^*\|_2 + \|\widehat{\boldsymbol{\theta}}_T - \boldsymbol{\theta}_T^*\|_2 \leq \frac{9}{\rho_{\min}} \max\left\{\sqrt{d_x}\lambda_{x,n}, \sqrt{d_y}\lambda_{y,n}\right\}\right) \geq 1 - c_1 p'^{-2},$$

*for some constant $c_1 > 0$.*

**Lemma 3.** *If $\frac{\max\left\{d_x\lambda_{x,n}^2, d_y\lambda_{y,n}^2\right\}}{\min\{\lambda_{x,n}, \lambda_{y,n}\}} \leq \frac{\rho_{\min}^2}{1296\rho_{\max}\kappa_3(n,p)\log p'} \frac{\alpha}{2-\alpha}, \quad \sqrt{d_x + d_y} \max\left\{\sqrt{d_x}\lambda_{x,n}, \sqrt{d_y}\lambda_{y,n}\right\} \leq \frac{\rho_{\min}^2}{40\rho_{\max}\kappa_3(n,p)\log p'}$, and $\|W^n\|_\infty \leq \frac{\lambda_n}{4}$, then we have*

$$P\left(\frac{\|R^n\|_\infty}{\min\{\lambda_{x,n}, \lambda_{y,n}\}} \leq \frac{\alpha}{4(2-\alpha)}\right) \geq 1 - c_1 p'^{-2},$$

*for some constant $c_1 > 0$.*

Armed with these lemmas, the proof of Theorem 3 is straightforward: Consider the choice of regularization parameters

$$\lambda_{x,n} = \frac{8(2-\alpha)}{\alpha}\sqrt{\kappa_1(n,p)\kappa_4}\sqrt{\frac{\log q}{n}} \quad, \text{ and } \quad \lambda_{y,n} = \frac{8(2-\alpha)}{\alpha}\sqrt{\kappa_1(n,p)\kappa_4}\sqrt{\frac{\log p}{n}}.$$

Then for $n \geq \max\left\{\frac{4}{\kappa_1(n,p)\kappa_2(n,p)^2\kappa_4}, \frac{16\kappa_h^2}{\kappa_4^2}\right\}\log p'$, the conditions of Lemma 1 are satisfied, hence (25) holds with high probability. Moreover, given (25) holds, with a sufficiently large sample size $n \geq L'\left(\frac{2-\alpha}{\alpha}\right)^4 (d_x + d_y)^2\kappa_1(n,p)\kappa_3(n,p)^2(\log p + \log q)(\log p')^2$ for some constant $L' > 0$, the conditions of Lemma 2 and 3 are also satisfied, and therefore, the resulting statements in Lemma 2 and 3 also hold with high probability.

*Strict dual feasibility.* By some algebra, we obtain

$$\lambda_{x,n}\widehat{Z}^x_{T^c} = Q^*_{T^cT}(Q^*_{TT})^{-1}[-W^n_T + R^n_T - \lambda_{x,n}\widehat{Z}^x_T] + W^n_{T^c} - R^n_{T^c}$$
$$\lambda_{y,n}\widehat{Z}^y_{S^c} = Q^*_{S^cS}(Q^*_{SS})^{-1}[-W^n_S + R^n_S - \lambda_{y,n}\widehat{Z}^y_S] + W^n_{S^c} - R^n_{S^c}.$$

Therefore, by Hölder's inequality and the fact that $\|\widehat{Z}^y_S\|_\infty \leq 1$,

$$\|\widehat{Z}^y_{S^c}\|_\infty \leq \|Q^*_{S^cS}(Q^*_{SS})^{-1}\|_\infty\Big[\frac{\|W^n_S\|_\infty}{\lambda_{y,n}} + \frac{\|R^n_S\|_\infty}{\lambda_{y,n}} + 1\Big] + \frac{\|W^n_{S^c}\|_\infty}{\lambda_{y,n}} + \frac{\|R^n_{S^c}\|_\infty}{\lambda_{y,n}}$$

$$\leq (1-\alpha) + (2-\alpha)\Big[\frac{\|W^n_y\|_\infty}{\lambda_{y,n}} + \frac{\|R^n\|_\infty}{\lambda_{y,n}}\Big]$$

$$\leq (1-\alpha) + (2-\alpha)\Big[\frac{\|W^n_y\|_\infty}{\lambda_{y,n}} + \frac{\|R^n\|_\infty}{\min\{\lambda_{x,n},\lambda_{y,n}\}}\Big] \leq (1-\alpha) + \frac{\alpha}{4} + \frac{\alpha}{4} = 1 - \frac{\alpha}{2} < 1.$$

Similarly, we have

$$\|\widehat{Z}^x_{T^c}\|_\infty \leq (1-\alpha) + (2-\alpha)\Big[\frac{\|W^n_x\|_\infty}{\lambda_{x,n}} + \frac{\|R^n\|_\infty}{\min\{\lambda_{x,n},\lambda_{y,n}\}}\Big] \leq (1-\alpha) + \frac{\alpha}{4} + \frac{\alpha}{4} = 1 - \frac{\alpha}{2} < 1.$$

*Correct sign recovery.* To guarantee that the support of $\widehat{\theta}$ is not strictly within the true support $S$, it suffices to show that $\max\left\{\|\widehat{\theta}_S - \theta^*_S\|_\infty, \|\widehat{\theta}_T - \theta^*_T\|_\infty\right\} \leq \frac{\theta^*_{\min}}{2}$. From Lemma 2, we have

$$\max\left\{\|\widehat{\theta}_S - \theta^*_S\|_\infty, \|\widehat{\theta}_T - \theta^*_T\|_\infty\right\} \leq \|\widehat{\theta}_S - \theta^*_S\|_2 + \|\widehat{\theta}_T - \theta^*_T\|_2$$
$$\leq \frac{5}{\rho_{\min}}\max\left\{\sqrt{d_x}\lambda_{x,n}, \sqrt{d_y}\lambda_{y,n}\right\} \leq \frac{\theta^*_{\min}}{2}$$

as long as $\theta^*_{\min} \geq \frac{10}{\rho_{\min}}\max\left\{\sqrt{d_x}\lambda_{x,n}, \sqrt{d_y}\lambda_{y,n}\right\}$, which completes the proof.

### B.3   Proof of Lemma 1

For the proof, we first define two events that would be useful even in the proofs of the remaining lemmas:

$$\xi_1 := \left[\max_{i,s,u}\left\{|B_s(Y^{(i)}_s)|, |B_u(X^{(i)}_u)|\right\} \leq 4\log p'\right] \text{ and}$$

$$\xi_2 := \left[\max_{s,u}\left\{\frac{1}{n}\sum_{i=1}^n B_s\big(Y^{(i)}_s\big)^2, \frac{1}{n}\sum_{i=1}^n B_u\big(X^{(i)}_u\big)^2\right\} \leq \kappa_4\right].$$

Then, by Proposition 1, the probabilities with which each event occurs are at least

$$P[\xi^c_1] \leq c_1\,n(p+q)p'^{-4} \leq c_1\,p'^{-2},$$

$$P[\xi^c_2] \leq \exp\big(-\frac{\kappa_4^2}{4\kappa_h^2}n + \log(p+q)\big) \leq \exp(-c_2 n),$$

as long as $n \geq \frac{8\kappa_h^2}{\kappa_4^2}\log(p+q)$.

Now, for a fixed $t \in V \backslash s$, we define $V_t^{(i)}$ for notational convenience so that

$$W_t^n = \frac{1}{n}\sum_{i=1}^n B_s(Y_s^{(i)})B_t(Y_t^{(i)}) - B_t(Y_t^{(i)})D'\Big(\theta_s^* + \sum_{u\in V'}\theta_{su}^*B_u(X_u) + \sum_{t\in V\backslash s}\theta_{st}^*B_t(Y_t)\Big) = \frac{1}{n}\sum_{i=1}^n V_t^{(i)}.$$

Conditioned on the events $\xi_1$ and $\xi_2$, by the definition of the moment generating function and standard Chernoff bound technique, we obtain

$$P\Big[\frac{1}{n}\sum_{i=1}^n |V_t^{(i)}| > \frac{\alpha}{2-\alpha}\frac{\lambda_n}{4} \mid \xi_1, \xi_2\Big] \le 2\exp\Big(-\frac{\alpha^2}{(2-\alpha)^2}\frac{n\lambda_{y,n}^2}{32\kappa_1(n,p)\kappa_4}\Big),$$

as long as $\frac{\alpha}{2-\alpha}\frac{\lambda_{y,n}}{4} \le \kappa_1(n,p)\kappa_2(n,p)\kappa_4$ for large enough $n$ (For details, see the proof of Lemma 2 in [10]). By a union bound over $V \backslash s$, we obtain

$$P\Big[\|W_y^n\|_\infty > \frac{\alpha}{2-\alpha}\frac{\lambda_n}{4} \mid \xi_1, \xi_2\Big] \le 2\exp\Big(-\frac{\alpha^2}{(2-\alpha)^2}\frac{n\lambda_{y,n}^2}{32\kappa_1(n,p)\kappa_4} + \log p\Big).$$

Therefore, provided that $\lambda_{y,n} \ge \frac{8(2-\alpha)}{\alpha}\sqrt{\kappa_1(n,p)\kappa_4}\sqrt{\frac{\log p}{n}}$, we obtain

$$P\Big[\|W_y^n\|_\infty > \frac{\alpha}{2-\alpha}\frac{\lambda_{y,n}}{4} \mid \xi_1, \xi_2\Big] \le \exp(-c_3'n).$$

By a very similar process for a set $V'$, we have

$$P\Big[\|W_x^n\|_\infty > \frac{\alpha}{2-\alpha}\frac{\lambda_{x,n}}{4} \mid \xi_1, \xi_2\Big] \le \exp(-c_3'n),$$

for a $\lambda_{x,n} \ge \frac{8(2-\alpha)}{\alpha}\sqrt{\kappa_1(n,p)\kappa_4}\sqrt{\frac{\log q}{n}}$. Finally, we have the resulting statement in the lemma by utilizing the fact that $P(A_1 \text{ or } A_2) \le P(\xi_1^c) + P(\xi_2^c) + P(A_1|\xi_1, \xi_2) + P(A_2|\xi_1, \xi_2)$.

## B.4 Proof of Lemma 2

In order to establish the error bound $\|\widehat{\boldsymbol\theta}_S - \boldsymbol\theta_S^*\|_2 + \|\widehat{\boldsymbol\theta}_T - \boldsymbol\theta_T^*\|_2 \le B$ for some radius $B$, we can extend the results in the several previous works (e.g. [26, 13]) and prove that it suffices to show $F(u_T, u_S) > 0$ for all $u_T := \boldsymbol\theta_T - \boldsymbol\theta_T^*$ and $u_S := \boldsymbol\theta_S - \boldsymbol\theta_S^*$ s.t. $\|u_T\|_2 + \|u_S\|_2 = B$ where

$$F(u_T, u_S) := \ell(\boldsymbol\theta_T^* + u_T, \boldsymbol\theta_S^* + u_S; \mathcal{Z}) - \ell(\boldsymbol\theta_T^*, \boldsymbol\theta_S^*; \mathcal{Z})$$
$$+ \lambda_{x,n}(\|\boldsymbol\theta_T^* + u_T\|_1 - \|\boldsymbol\theta_T^*\|_1) + \lambda_{y,n}(\|\boldsymbol\theta_S^* + u_S\|_1 - \|\boldsymbol\theta_S^*\|_1).$$

Note again that $T$ is the true support set of $\boldsymbol\theta^x$ and $S$ is that of $\boldsymbol\theta^y$. Note also that for $\hat{u}_T := \widehat{\boldsymbol\theta}_T - \theta_T^*$ and $\hat{u}_S := \widehat{\boldsymbol\theta}_S - \theta_S^*$, $F(\hat{u}_T, \hat{u}_S) \le 0$ and $F(0,0) = 0$. Below we show that $F(u_T, u_S)$ is strictly positive on the boundary of the ball with radius $B = M\max\{\sqrt{d_x}\lambda_{x,n}, \sqrt{d_y}\lambda_{y,n}\}$ where $M > 0$ is a parameter that we will choose later in this proof.

Some algebra yields

$$F(u_T, u_S) \ge \Big(\max\{\sqrt{d_x}\lambda_{x,n}, \sqrt{d_y}\lambda_{y,n}\}\Big)^2\Big\{-\frac{1}{4}M + q^*M^2 - 2M\Big\} \tag{26}$$

where $q^*$ is the minimum eigenvalue of $\nabla^2\ell(\boldsymbol\theta_T^* + vu_T, \boldsymbol\theta_S^* + vu_S; \mathcal{Z})$ for some $v \in [0,1]$. Moreover, by the similar reasoning as in the case of Lemma 3 of [10], we can find the lower bound of $q^*$:

$$q^* \ge \rho_{\min} - 4\rho_{\max}M\sqrt{d_x + d_y}\max\{\sqrt{d_x}\lambda_{x,n}, \sqrt{d_y}\lambda_{y,n}\}\kappa_3(n,p)\log p',$$

conditioned on $\xi_1$. From (26), we obtain

$$F(u_T, u_S) \ge (\lambda_n\sqrt{d})^2\Big\{-\frac{1}{4}M + \frac{\rho_{\min}}{2}M^2 - 2M\Big\},$$

as long as $\sqrt{d_x + d_y}\max\{\sqrt{d_x}\lambda_{x,n}, \sqrt{d_y}\lambda_{y,n}\} \le \frac{\rho_{\min}}{8\rho_{\max}M\kappa_3(n,p)\log p'}$.

Finally, we set $M = \frac{9}{\rho_{\min}}$ so that $F(u_T, u_S)$ is strictly positive, and hence we can conclude that

$$\|\widehat{\boldsymbol\theta}_S - \boldsymbol\theta_S^*\|_2 + \|\widehat{\boldsymbol\theta}_T - \boldsymbol\theta_T^*\|_2 \le \frac{9}{\rho_{\min}}\max\{\sqrt{d_x}\lambda_{x,n}, \sqrt{d_y}\lambda_{y,n}\},$$

provided that $\sqrt{d_x + d_y}\max\{\sqrt{d_x}\lambda_{x,n}, \sqrt{d_y}\lambda_{y,n}\} \le \frac{\rho_{\min}^2}{72\rho_{\max}\kappa_3(n,p)\log p'}$.

## B.5  Proof of Lemma 3

Again from the similar reasoning as in the proof of Lemma 4 of [10], we have

$$|R_t^n| \leq 4\kappa_3(n,p)\rho_{\max}\log p' \|\widehat{\boldsymbol{\theta}}_{T;S} - \boldsymbol{\theta}_{T;S}^*\|_2^2 \leq 4\kappa_3(n,p)\rho_{\max}\log p' \left(\|\widehat{\boldsymbol{\theta}}_S - \boldsymbol{\theta}_S^*\|_2 + \|\widehat{\boldsymbol{\theta}}_T - \boldsymbol{\theta}_T^*\|_2\right)^2$$

for all $t \in V \backslash s\{1,...,p-1\} \cup V'$. Therefore, if Lemma 2 holds, then

$$\|R^n\|_\infty \leq \frac{324\rho_{\max}\kappa_3(n,p)\log p'}{\rho_{\min}^2} \max\left\{d_x\lambda_{x,n}^2, d_y\lambda_{y,n}^2\right\}$$

which is equivalent with

$$\frac{\|R^n\|_\infty}{\min\{\lambda_{x,n}, \lambda_{y,n}\}} \leq \frac{324\rho_{\max}\kappa_3(n,p)\log p'}{\rho_{\min}^2} \frac{\max\left\{d_x\lambda_{x,n}^2, d_y\lambda_{y,n}^2\right\}}{\min\{\lambda_{x,n}, \lambda_{y,n}\}} \leq \frac{\alpha}{4(2-\alpha)}$$

by the assumption of the lemma.