[Reviews · NeurIPS 2013]

Submitted by Assigned_Reviewer_4

This paper extends the work of Yang et al (2012) on undirected graphical models where the node-wise conditional distributions are from the univariate exponential family; in this paper, the authors allow covariates to be included and also allow the distribution type to be node specific. They derive the form of the "natural parameter" for the response distributions under the assumption that the node-wise conditional distributions of the covariates are also from the univariate exponential family, and prove guarantees on the learnability of the models. Finally, the authors present some limited experimental results on synthetic and real data.

The paper is well written, particularly given the density of material. I found only one typo (page 3, "A(\theta(X) > " is missing a parenthesis). To the best of my knowledge this is original.

The paper is certainly of theoretical interest, and the experiments suggest they may have practical impact as well. I'm curious as to whether allowing different distribution types will lead to practical advances.

It took me quite a while to digest Equation (2). What threw me at first was the indexing {t_2, ..., t_k} in the last sum, which I thought meant we were considering all subsets except for the full N(s) - it makes sense now, but that did slow me down. Also, can't the last term there "absorb" all of the previous ones? For degree 2, it maybe makes some sense to have both node terms and edge terms because you might think that evidence for a state and transition have some semantic difference, but here it seems to just lengthen the equation.
Summary: Well written paper that extends existing results to allow covariates.

Submitted by Assigned_Reviewer_6

This interesting paper contributes to the literature on CRFs with covariate-dependent structure, making three separate and interesting contributions:

(a) (Theorem 1) generalizes the classical and well-known correspondence between autoregressive Gaussian models and multivariate Gaussians
to arbitrary. The interesting and subtle bit here is the dependence of equations (2) and (3) on X. We started in (1) by allowing the autoregressive
weights to depend arbitrarily on X and Y\s, but by making an additional CRF assumption on P(Y|X) we find that these weights factorize
into a terms that depends on X and the sufficient statistics of Y_s's neighbours.

(b) Section 2.1: If we assume an exponential family and graphical model over X in addition to our CRF P(Y|X), this implies additional structure in the model parameters of the CRF. This is a neat idea.

(c) The structure from (b) allows one to use a particular l1 regularizer for structure discovery, for which we can provide guarantees about learning error and structure recovery.

Although (a) and (c) do appear to follow relatively closely the previous results from Yang et al about MRFs, (b) certainly seems more new,
as it is specific to conditional models.

One (perhaps too ambitious) question that I had was regarding the marginal distribution P(Y_s | X). If the node conditionals are Gaussian, then of course these marginals are Gaussian; similarly if the node conditions are fully-parameterized discrete distributions, then the node marginals are also in the same exponential family. I don't suppose that this property would hold for other exponential families?

I have only minor suggestions for improvement:

* I'm not sure that the Introduction sells the contribution of the paper as clearly as it could, as it seems to focus on having different exponential families in the autoregressive structure, where it seems to me the real modelling contribution is the interplay among the autoregressive weights and the sufficient statistics of X.

* The real data example in Section 4 seems like it is probably a nice example of the modelling benefits of having covariate-dependent structure, although this is almost impossible to understand for readers who are not already familiar with the biology in this example. It would be nice to expand the discussion of these results for readers coming from a more general ML background, althought I understand that this could be difficult given space constraints.

* Theorem (1): The statement of the theorem is fairly long, and the phrase "These assertions are consistent" is a bit less precise than I'd perhaps like.
I wonder if it's possible to introduce some terminology that makes clauses (a) and (b) shorter and therefore the theorem easier to understand.

* Equation (2): I am assuming that if the functions B_s return a vector, then the product should be interpreted elementwise?

* line 157: B_s(Y_s) = Y_s. Surely Y_s^2 needs to be a sufficient statistic somehow?

* line 298: This is a *pseudo*-likelihood, yes?

* line 316: Are the constants M_1 and M_2 computable? Is there a simple expression for them? Are these constraints always satisfiable?
Summary: This is a nice paper that contributes to the literature on CRFs with covariate-dependent structure.

Submitted by Assigned_Reviewer_8

This paper extends the results in Yang et al "Graphical models via generalized linear models", from NIPS 2012, to conditional random fields as well as Markov random fields. The main modeling idea is to specify a conditional random field by specifying the node-conditional distributions (that is, P(Y_s | Y_{\s},X) ) of the response variables Y. If these node-conditional distributions factorize as functions of outer products of sufficient statistics of the response variables Y then it is possible to represent the full joint distribution in a similar form.

Theorem 1 shows how to write down the full joint distribution generally, and theorem 2 shows how to generalize the standard correspondence between generative and discriminative models to this setting. It is odd that theorem 2 is a theorem since it is stated without proof, as a proof is not necessary because it seems to follow from the discussion preceding it.

Theorem 3 generalizes the structure learning results of Yang et al to this setting cleanly. The proofs seem to be very similar to those in Yang et al.

There is some experimental validation that the model seems to be capable of representing the distributions of interest. However, the supplemental material could have some of the plots and figures which were omitted due to lack of space.
Summary: The paper presents a new way of expressing conditional random fields with continuous variables, and shows how to do structure learning in that setting. It appears to be correct, but its contributions are relatively incremental.
Author Feedback

Author rebuttal: We are very thankful to the reviewers for their careful and encouraging comments and feedback.

Reviewer 4: It is an interesting point for further research whether the exponential family in Theorem~1 could be expressed with more compact parameters. Note however that the last term in Theorem~1 may not necessarily be able to absorb the previous terms; in particular the sufficient statistics of different orders could be linearly independent. (They can be absorbed if the higher order sufficient statistic was an arbitrary function of the comprising response variables, but here the sufficient statistics have a specific tensor form).


Reviewer 6: "One (perhaps too ambitious) question that I had was regarding the marginal distribution P(Y_s | X). If the node conditionals are Gaussian, then of course these marginals are Gaussian; similarly if the node conditions are fully- parameterized discrete distributions, then the node marginals are also in the same exponential family. I don't suppose that this property would hold for other exponential families?"

This is a very salient question; but it need not always hold; it doesn't hold for the Poisson case for instance.

On Equation (2): In this work, we were focusing only on the case where the sufficient statistics B is a scalar. For future work, we are considering extending it to allow for vector sufficient statistics.

On line 157: B(Y) is linear in all our examples (Gaussian, Poisson, exponential and so on). In the Gaussian case, the square term arises through the base measure, C().

On line 298: It is the log-likelihood of 'conditional' distribution. We learn the CRF graph structure by estimating node-neighborhoods independently (by optimizing regularized conditional log-likelihoods), and combine them by 'and' or 'or' rules. This is closely related to the joint estimator that optimizes the pseudolikelihood.

On line 316: The detailed description of M_1 and M_2 is provided in the appendix (in Lemma 1). They do have a simple expression; they are constants depending only on the incoherence condition and the log partition function of node-conditional distribution.


Reviewer 8: "It is odd that theorem 2 is a theorem since it is stated without proof, as a proof is not necessary because it seems to follow from the discussion preceding it."

Theorem 2 specifies the form of P(Y|X) under the additional assumption on P(X_u | X_\u , Y) in (6). As such it is an important result since as Reviewer_6 points out it specifies the interplay between the response and the covariates in the covariate feature functions in CRFs. Given the need for specifying such feature functions in practical applications of CRFs, this would hopefully add to the off-the-shelf CRF modeling toolkit. The proof is natural to the extent that it follows the lines of the proof of Theorem 1; and was omitted, but we will be glad to add it to the appendix.